## [Transparent Peer Review file · Nature Communications]

Identification and structural characterization of anthrax toxin receptor 2 as the *Clostridium perfringens* NetF receptor

Corresponding Author: Professor Benoît Zuber

Version 0:

Reviewer comments:

Reviewer #1

(Remarks to the Author)

In this study, Wang and colleagues have described the pore-forming toxin NetF, a member of the beta-PFT family and identified its cellular receptor, ANTXR2. The structures of NetF in the pre-pore and pore conformation were solved to high resolution using Cryo-EM as well as the structure of the pore form in complex with the extracellular domain of ANTXR2. In addition, the authors have characterised in much detail, the binding properties of the receptor and identified the part of the receptor responsible for cell susceptibility. The paper is very well written and contains numerous controls that validate the authors' conclusions. Furthermore, the structure of the complex is one of a few available describing the detailed molecular structure of a toxin and its receptor and is the first structure of the soluble part of the ANTXR2 receptor in the literature.

In my opinion, this study justifies publication in Nature Communications. Besides being an elegant and very complete study of the system, it provides important insights into the mode of action of these beta-PFTs and paves the way to design molecules to inhibit binding to susceptible cells. It would be good however, for the authors to clarify the points below for the readers:

1. The authors describe seeing binding and oligomerisation on U937 wt and resistant cell lines as well as the KO cell lines to similar levels. This observation is very interesting and implies that oligomerisation itself, is not a pre-requisite for pore formation. However, the percentage of oligomers compared to monomer is very little. It would be useful for the authors to comment on this. Is the majority of monomer seen due to oversaturation of the cells? For example, in Sup.Fig1B the majority of NetF has oligomerised on liposomes with cholesterol in the absence of a receptor and has presumably formed pores since they go on to solve the pore structure from these particles.

2. The extra densities seen for the N-terminus and suggested by the authors to be the His-tag is somewhat speculative and weakens an otherwise solid study. Of course, a conclusive experiment here would be to cleave the His-tag and see if this density disappears. In the absence of this experiment, I don't think one cannot draw conclusions as to the location of the N-terminus and the text should be more indicative of the presumption taken here. In addition, in Figure 2A, a ring of flat density, presumably lipid, is shown below the pre-pore but not mentioned in the text. Is this noise from the nanodisc?

3. In figure 5B, the map is described as a local resolution map? Should that read local resolution filtered map as the colouring is not indicative of local resolution?

Reviewer #2

(Remarks to the Author)

This study provides remarkable structural evidence on the pore formation and binding of NetF, belonging to the beta-haemolysin family. There are few limitations regarding the cell specificity of NetF that require clarification which could be substantiated by necessary controls.

a) More information on the toxicity kinetic would complete the information provided in Figures 3a and 3b. 24h incubation with NetF is not informative enough to grasp the extent of NetF toxicity activity. A kinetic is required to provide more information on the dynamic of NetF toxicity, and to better introduce the notion of specificity to ANTXR2

b) Additional information should be provided to support the Figures 3a and 3b. Figure 3f shows that the authors have used

an antibody targeted against endogenous ANTRX2.

Although the authors have shown that ectopic human ANTRX2 expression in HAP1 cells (but not murine) render them sensitive to NetF toxicity, these data should be further completed with evidence showing the levels of ANTRX2 in the tested cell lines. The same antibody used in Figure 3f should be used against the cell lines tested in Figures 3a and 3b to substantiate and/or corroborate their insensitivity to NetF.

c) In Figure 3c and 3d, the authors have used a CRISPR-Cas9 strategy to ablate ANTRX2 in U937 cells. In addition to sequencing results shown in Figure 3c, a western-blot using the ANTRX2 antibody should be shown to support the further support the Figure 3c and 3d data.

Reviewer #3

(Remarks to the Author)
Review Report

This manuscript presents a comprehensive study combining structural biology and cellular approaches to identify and characterize ANTRX2 (also known as CMG2) as the receptor for NetF, a β -pore-forming toxin (β PFT) from *Clostridium perfringens*. The authors provide high-resolution cryo-EM structures of NetF in both pre-pore and pore conformations, as well as in complex with the PMT-modified extracellular domain of ANTRX2. They demonstrate that NetF engages ANTRX2 through a unique binding mode involving primarily the Ig-like domain but also the vWA domain, distinct from the interaction of anthrax protective antigen (PA) with the same receptor. Functional assays, including CRISPR-Cas9 mutagenesis and receptor ortholog screening, support the conclusion that ANTRX2 is essential for NetF-mediated cytotoxicity.

Major comment:

The authors convincingly demonstrate the importance of the Ig-like domain of ANTRX2 in NetF binding and toxicity. However, the mutagenesis data (i.e., in-frame mutations from the CRISPR-Cas9 experiments) should be summarized in a dedicated table, along with interaction data between ANTRX2 and NetF. This would clarify the correlation between mutation types and resistance phenotypes, especially in light of the structural findings that highlight the Ig domain as a key interaction site. A table listing key amino acid residues involved in the ANTRX2–NetF interaction should also be provided. Furthermore, the conservation of critical residues in the vWA domain across species should be discussed in relation to the observed resistance of murine ANTRX2-expressing cells. It is important to discuss how ANTRX2 and NetF interaction data correlate between mutagenesis, structure, and species-specific variants of ANTRX2.

Other comments:

Point 1 : The intracellular retention of canine ANTRX2 in HAP1 cells is shown, but its membrane localization in HEK293 cells is less evident. The authors should perform plasma membrane fractionation or surface biotinylation assays to confirm surface expression levels. Additionally, a non-transfected HAP1 control is missing in the relevant figure.

Point 2 (Figure 4A and 4C) : There are noticeable differences in ANTRX2 expression levels across species. The authors should discuss whether these differences could influence the observed susceptibility to NetF, particularly for murine ANTRX2, which is expressed at lower levels. Moreover, given the low membrane expression of cANTRX2 and its inverse correlation with high cell susceptibility to NetF, the lack of correlation between expression levels and susceptibility should be addressed, as this is unexpected for a toxin receptor.

Point 3 (Figure 4D–4E): The chimeric receptor experiments highlight the importance of the Ig-like domain. However, the cryo-EM structure also implicates specific residues in the vWA domain. The authors should discuss whether these residues are conserved between human and murine ANTRX2.

Point 4 : The comparison between NetF and anthrax PA is somewhat speculative. While both toxins use ANTRX2, their mechanisms of action and physiological effects differ significantly. This section should be shortened and more cautiously framed.

Point 5 (Sup. Fig. 1B) : provide quantifications showing that cholesterol increases membrane binding of NetF.

Overall, this is a well-executed and thorough study that significantly advances our understanding of clostridial β PFTs and their receptor interactions. The structural and functional data are robust and well-integrated. Addressing the major comment and clarifying the minor points will further strengthen the manuscript.

Version 1:

Reviewer comments:

Reviewer #1

(Remarks to the Author)

The revised version of the manuscript has dealt with my comments sufficiently and performed additional experiments to

support the hypothesis. I have no further comments and support the publication in Nature Communications.

Reviewer #2

(Remarks to the Author)

The authors have provided a commendable revision. Well done all.

Reviewer #3

(Remarks to the Author)

The authors responded positively to all of my comments

Point-by-point answer to reviewers comments

Reviewer #1 (Remarks to the Author):

In this study, Wang and colleagues have described the pore-forming toxin NetF, a member of the beta-PFT family and identified its cellular receptor, ANTXR2. The structures of NetF in the pre-pore and pore conformation were solved to high resolution using Cryo-EM as well as the structure of the pore form in complex with the extracellular domain of ANTXR2. In addition, the authors have characterised in much detail, the binding properties of the receptor and identified the part of the receptor responsible for cell susceptibility. The paper is very well written and contains numerous controls that validate the authors conclusions. Furthermore, the structure of the complex is one of a few available describing the detailed molecular structure of a toxin and its receptor and is the first structure of the soluble part of the ANTXR2 receptor in the literature.

In my opinion, this study justifies publication in Nature Communications. Besides being an elegant and very complete study of the system, it provides important insights into the mode of action of these beta-PFTs and paves the way to design molecules to inhibit binding to susceptible cells. It would be good however, for the authors to clarify the points below for the readers:

We are grateful for the reviewer positive feedback and constructive comments.

1. The authors describe seeing binding and oligomerisation on U937 wt and resistant cells lines as well as the KO cell lines to similar levels. This observation is very interesting and implies that oligomerisation itself, is not a pre-requisite for pore formation. However, the percentage of oligomers compared to monomer is very little. It would be useful for the authors to comment on this. Is the majority of monomer seen due to oversaturation of the cells? For example, in Sup.Fig1B the majority of NetF has oligomerised on liposomes with cholesterol in the absence of a receptor and has presumably formed pores since they go on to solve the pore structure from these particles.

This is an interesting observation. NetF can efficiently oligomerize on liposomes, particularly when they contain a high fraction of cholesterol (see revised Sup. Fig. 1). In contrast to liposomes, approximately 50% of the mass of plasma membranes consist of proteins (Alberts et al., 2022, *Molecular biology of the cell*, 7th ed.). We speculate that this high protein content may hinder NetF access to, and insertion into the membrane, and that a receptor could serve to dock the forming oligomer. In the absence of a receptor it is possible that some NetF molecules form oligomers at the cell surface, but these may detach before their beta-barrel can insert into the membrane. An experimental validation of this hypothesis is beyond the scope of the present study but we mention it in the Discussion.

2. The extra densities seen for the N-terminus and suggested by the authors to be the His-tag is somewhat speculative and weakens an otherwise solid study. Of course, a conclusive experiment here would be to cleave the his-tag and see if this density disappears. In the absence of this experiment, I don't think one cannot draw conclusions as to the location of the N-terminus and the text should be more

indicative of the presumption taken here. In addition, in Figure 2A, a ring of flat density, presumably lipid, is shown below the pre-pore but not mentioned in the text. Is this noise from the nanodisc?

We performed focused classification on the membrane-distal region of the pre-pore and pore datasets. The backbone of some additional N-terminal residues is now resolved (Figure 2D, red density). A clear conformational change can be observed: in the pre-pore, the residues protrude from the vestibule, whereas in the pore they form part of its inner surface. The most N-terminal residues remain unresolved, suggesting that this region is highly flexible. The ring of flat density below the pre-pore indeed originates from the nanodisc, as confirmed by the image. We now mention it in the figure legend.

3. In figure 5B, the map is described as a local resolution map? Should that read local resolution filtered map as the colouring is not indicative of local resolution?

We thank the reviewer for noticing this. This was a mistake and it has been corrected. The coloring indicates the different domains rather than the local resolution.

Reviewer #2 (Remarks to the Author):

This study provides remarkable structural evidence on the pore formation and binding of NetF, belonging to the beta-haemolysin family. There are few limitations regarding the cell specificity of NetF that require clarification which could be substantiated by necessary controls.

We thank the reviewer for the encouraging feedback and helpful remarks.

a) More information on the toxicity kinetic would complete the information provided in Figures 3a and 3b. 24h incubation with NetF is not informative enough to grasp the extent of NetF toxicity activity. A kinetic is required to provide more information on the dynamic of NetF toxicity, and to better introduce the notion of specificity to ANTRX2

We performed the requested experiment and it is now included in Sup. Fig. 4 A and B.

b) Additional information should be provided to support the Figures 3a and 3b. Figure 3f shows that the authors have used an antibody targeted against endogenous ANTRX2.

Although the authors have shown that ectopic human ANTRX2 expression in HAP1 cells (but not murine) render them sensitive to NetF toxicity, these data should be further completed with evidence showing the levels of ANTRX2 in the tested cell lines. The same antibody used in Figure 3f should be used against the cell lines tested in Figures 3a and 3b to substantiate and/or corroborate their insensitivity to NetF.

This is a valid point, and we have worked to address it, testing many primary antibodies. Unfortunately, antibody-based detection of endogenous ANTRX2 protein is extremely challenging, as confirmed by the long-standing experience of coauthors

who have worked on this protein for two decades (LA and FGvdG). To complement these efforts, we quantified ANTXR2 mRNA levels by qRT-PCR (Sup. Fig. 4E). Overall, transcript abundance correlated with the sensitivity of cell lines to NetF, with the exception of murine lines. This is consistent with our observation that ectopic expression of murine ANTXR2 does not render cells susceptible to NetF. The canine fibroblast line that we had originally established was unfortunately no longer available at the time of revision. Because we were therefore unable to perform a reliable quantification of ANTXR2 mRNA levels in this line, we chose to remove the corresponding data from the manuscript to avoid presenting incomplete or non-verifiable results.

c) In Figure 3c and 3d, the authors have used a CRISPR-Cas9 strategy to ablate ANTRX2 in U937 cells. In addition to sequencing results shown in Figure 3c, a western blot using the ANTRX2 antibody should be shown to support the further support the Figure 3c and 3d data.

As noted above, endogenous ANTXR2 cannot be detected by Western blot. Therefore, we could not perform this additional control.

Reviewer #3 (Remarks to the Author):

Review Report

This manuscript presents a comprehensive study combining structural biology and cellular approaches to identify and characterize ANTXR2 (also known as CMG2) as the receptor for NetF, a β -pore-forming toxin (β PFT) from *Clostridium perfringens*. The authors provide high-resolution cryo-EM structures of NetF in both pre-pore and pore conformations, as well as in complex with the PMT-modified extracellular domain of ANTXR2. They demonstrate that NetF engages ANTXR2 through a unique binding mode involving primarily the Ig-like domain but also the vWA domain, distinct from the interaction of anthrax protective antigen (PA) with the same receptor. Functional assays, including CRISPR-Cas9 mutagenesis and receptor ortholog screening, support the conclusion that ANTXR2 is essential for NetF-mediated cytotoxicity.

We thank the reviewer for their positive assessment.

Major comment:

The authors convincingly demonstrate the importance of the Ig-like domain of ANTXR2 in NetF binding and toxicity. However, the mutagenesis data (i.e., in-frame mutations from the CRISPR-Cas9 experiments) should be summarized in a dedicated table, along with interaction data between ANTXR2 and NetF. This would clarify the correlation between mutation types and resistance phenotypes, especially in light of the structural findings that highlight the Ig domain as a key interaction site. A table listing key amino acid residues involved in the ANTXR2–NetF interaction should also be provided. Furthermore, the conservation of critical residues in the vWA domain across species should be discussed in relation to the observed resistance of murine ANTXR2-expressing cells. It is important to discuss how ANTXR2 and NetF interaction data correlate between mutagenesis, structure, and

species-specific variants of ANTXR2.

We thank the reviewer for helpful suggestions. We have expanded Table S3, which originally showed residue pairs forming h-bonds and salt bridges. It now includes hydrophobic interactions as well. We have marked with a box all ANTXR2 residues directly interacting with NetF in ANTXR2 sequence in Sup. Fig. 7 A. Additionally this figure shows all the interaction sites that comprise a mutation in the murine sequence and their location with respect to vWA and Ig-like domains. We have added a new panel to Sup. Fig. 7 showing murine amino acids that are predicted by Alphafold3 to prevent complex formation with human ANTXR2. And we have added a discussion about this topic.

Other comments:

Point 1 : The intracellular retention of canine ANTXR2 in HAP1 cells is shown, but its membrane localization in HEK293 cells is less evident. The authors should perform plasma membrane fractionation or surface biotinylation assays to confirm surface expression levels. Additionally, a non-transfected HAP1 control is missing in the relevant figure.

A surface biotinylation assay is now included in Sup. Fig. 6C. A non-transfected HAP1 control has been added.

Point 2 (Figure 4A and 4C) : There are noticeable differences in ANTXR2 expression levels across species. The authors should discuss whether these differences could influence the observed susceptibility to NetF, particularly for murine ANTXR2, which is expressed at lower levels. Moreover, given the low membrane expression of cANTXR2 and its inverse correlation with high cell susceptibility to NetF, the lack of correlation between expression levels and susceptibility should be addressed, as this is unexpected for a toxin receptor.

Figure 4A shows expression level of overexpressed ANTXR2 variants in HAP1 cells and the associated viability is shown in Figure 4B. Note that Figure 4C shows cells viability in another cell line. In HAP1 cells, cANTXR2 is expressed to low levels and is not properly located at the plasma membrane (Sup. Fig. 6A). Consistently, cANTXR2 does not make HAP1 cells susceptible to NetF (Figure 4B). Since cells endogenously expressing cANTXR2 are highly susceptible to NetF (Figure 3A, Sup. Fig. 4E), we assessed expression of ectopic cANTXR2 in HEK293 cells. Surface expression was improved (Sup. Fig. 6BC) and consistently led to NetF toxicity (Figure 4C). The low susceptibility to NetF given by mANTXR2 may be explained by a lower affinity for NetF due to mutations in two binding sites located in the Ig-like domain (Sup. Fig. 7A). Corresponding point mutations significantly reduce binding affinity according to AlphaFold predictions (Sup. Fig. 7B).

Point 3 (Figure 4D–4E): The chimeric receptor experiments highlight the importance of the Ig-like domain. However, the cryo-EM structure also implicates specific residues in the vWA domain. The authors should discuss whether these residues are conserved between human and murine ANTXR2.

We have now indicated and discussed the conservation of residues between human and murine ANTXR2. We have also predicted the impact of point mutations on the binding affinity (see our answer to previous reviewer)

Point 4 : The comparison between NetF and anthrax PA is somewhat speculative. While both toxins use ANTXR2, their mechanisms of action and physiological effects differ significantly. This section should be shortened and more cautiously framed.

We have carefully reviewed our discussion and found little to no speculation in the comparison between NetF and PA. We highlight the differences in binding mode. PA-ANTXR2 structure has been published. There is clearly only interaction with the most membrane-distal part of ANTXR2 VWA domain and PA prepore (Lacy et al 2004). The length difference of the respective beta-barrels is also well established based on our study and Jiang et al 2015 study. It is therefore clear that PA prepore has no interaction with either membrane lipids or ANTXR2 Ig-like domain.

Point 5 (Sup. Fig. 1B) : provide quantifications showing that cholesterol increases membrane binding of NetF.

We have now performed a quantification which is included in Sup. Fig 1.

Overall, this is a well-executed and thorough study that significantly advances our understanding of clostridial β PFTs and their receptor interactions. The structural and functional data are robust and well-integrated. Addressing the major comment and clarifying the minor points will further strengthen the manuscript.